# Unique Enhancement of the Whispering Gallery Mode in Hexagonal Microdisk Resonator Array with Embedded Ge Quantum Dots on Si

**DOI:** 10.3390/nano13182553

**Published:** 2023-09-13

**Authors:** Zhifang Zhang, Jia Yan, Zuoru Dong, Ningning Zhang, Peizong Chen, Kun Peng, Yanyan Zhu, Zhenyang Zhong, Zuimin Jiang

**Affiliations:** 1State Key Laboratory of Surface Physics, Department of Physics, Fudan University, Shanghai 200438, China; zfzhang94@163.com (Z.Z.); 19110190054@fudan.edu.cn (J.Y.); zrdong2021@163.com (Z.D.); 16110190037@fudan.edu.cn (N.Z.); 16110190016@fudan.edu.cn (P.C.); 15110190021@fudan.edu.cn (K.P.); 2Shanghai Microwave Technology Research Institute, Shanghai 200063, China; 3College of Mathematics and Physics, Shanghai University of Electric Power, Shanghai 200090, China; yyzhu@shiep.edu.cn

**Keywords:** Ge QDs, hexagonal microdisk, whispering gallery modes, PL enhancement

## Abstract

The coupling between the quantum dots (QDs) and silicon-based microdisk resonator facilitates enhancing the light–matter interaction for the novel silicon-based light source. However, the typical circular microdisks embedded with Ge QDs still have several issues, such as wide spectral bandwidth, difficult mode selection, and low waveguide coupling efficiency. Here, by a promising structural modification based on the mature nanosphere lithography (NSL), we fabricate a large area hexagonal microdisk array embedded with Ge QDs in order to enhance the near-infrared light emissions by a desired whispering gallery modes (WGMs). By comparing circular microdisks with comparable sizes, we found the unique photoluminescence enhancement effect of hexagonal microdisks for certain modes. We have confirmed the WGMs which are supported by the microdisks and the well-correlated polarized modes for each resonant peak observed in experiments through the Finite Difference Time Domain (FDTD) simulation. Furthermore, the unique enhancement of the TE_5,1_ mode in the hexagonal microdisk is comparatively analyzed through the simulation of optical field distribution in the cavity. The larger enhanced region of the optical field contains more effectively coupled QDs, which significantly enhances the PL intensity of Ge QDs. Our findings offer a promising strategy toward a distinctive optical cavity that enables promising mode manipulation and enhancement effects for large-scale, cost-effective photonic devices.

## 1. Introduction

The self-assembled germanium quantum dots (Ge QDs) formed in the Stranski–Krastanov (S–K) mode have been demonstrated to have photoluminescence (PL) wavelengths that are highly suitable for optical communications band [1,2,3]. However, Ge QDs still suffer from several intrinsic drawbacks, such as insufficient efficiency, weak spectrum purity, low directivity, and inadequate emission intensity [4,5]. Embedding Ge QDs into high-quality factor (*Q*) and small mode volume (*V*_mode_) optical microcavities, such as photonic crystal (PhC) [6,7] and microdisk cavity [8,9], can significantly enhance the spontaneous emission rate at the selected wavelength through the Purcell effect [10,11]. Among them, circular microdisks have been applied to enhance the luminescence of QDs in cavities due to their support of whispering gallery modes (WGMs) [12]. In order to reduce the mode number in circular microdisk, the cavity size needs to be scaled down. However, miniaturized devices necessitate a higher precision in device fabrication and a higher efficiency in near-field interactions in optical cavities. Therefore, it is meaningful to explore the alternative cavity configurations with high near-field interactions for emission enhancement [13,14]. On the other hand, it has been observed that the WGMs present in the circular microdisks generally emit omnidirectionally along the edge, which leads to inefficient control of light direction and decreases the coupling efficiency between the microdisks and waveguide [15]. Based on the aforementioned points, it is worthwhile to conduct an investigation into novel configurational cavities.

A simple solution is to adopt regular polygonal microdisks that break rotational symmetry, which significantly limits light paths and enables more effective directional emission [16,17]. Strong light coupling can be achieved through the straight cavity side wall, which is beneficial for achieving high optical waveguide coupling efficiency. So far, it has been demonstrated that the WGM is supported in typical polygonal microcavities such as triangles [18], squares [19], hexagonals [20], and octagonals [21]. In these microcavities, hexagonal microdisks with diversified modes such as WGM and quasi-WGM have attracted much attention because of their relatively high symmetry [22]. Semiclassical ray models have been introduced to describe the WGM and quasi-WGM in hexagonal microdisks [23,24]; however, these models are two-dimensional and still cannot elucidate the resonant modes in a thick three-dimensional microcavity in detail. Therefore, more systematic investigations are needed to fully comprehend the unique characteristics and potential applications of hexagonal microdisks, especially for structures with a large microdisk thickness. Furthermore, systematic investigations on GeSi hexagonal microdisks have rarely been carried out, although there are some reports of the optical characteristics of hexagonal microdisks based on GaN [21], ZnO [23], and GaAs [25] semiconductor materials.

Herein, GeSi hexagonal microdisk arrays embedded with Ge-rich QDs have been fabricated via the modified nanosphere lithography (NSL) method. The isotropic gas of the RIE creates an anisotropic etching on the PS template of hexagonal close-packed arrangement, which causes the surface etching rate of the polymer sphere to be significantly higher than that at the contact point between the PS spheres, ultimately leading to the generation of the hexagonal surface template [26,27]. Subsequently, the periodically arranged hexagonal Si nanorods with good homogeneity and verticality were formed by ICP of cycle gas etching. In PL measurements, several significantly enhanced resonant light emission peaks supported in the hexagonal microdisk in the range of 1.2–2 μm are observed. In addition, the properties of hexagonal and circular microdisks are compared via the experimental results and Finite Difference Time Domain (FDTD) simulation. The optical field distribution of the WGMs in the hexagonal and circular microdisks is found to be much different despite the similarity in the microdisk size. The two cavities of different configurations also reveal different vertical cavity emission enhancements. Our results demonstrate the potential of Ge QDs–GeSi hexagonal microdisks for applications in the field of monolithic optical-electronic integrated circuits (MOEICs) on silicon.

## 2. Experiment

The Si buffer layer, Ge_0.11_Si_0.89_ alloy layer, and five Ge QDs layers were grown on Si (100) substrate by a solid source molecular beam epitaxy (MBE). Before growth, the Si substrate was cleaned by the standard RCA method and passivated by the HF to form a hydrogen-terminated surface. After thermal desorption at 860 °C, a 100 nm thick Si buffer layer was grown on Si substrate at the rate of 0.9 Å s^–1^ at 550 °C. The 100 nm thick Ge_0.11_Si_0.89_ alloy was deposited at 400 °C and the 5 layers of Ge QDs of ~8.6 Å (about 6.1 monolayers (MLs)) were grown in S–K mode at 520 °C. Figure 1a shows the typical surface topography after 8.6 Å Ge was deposited at 520 °C on the alloy Ge_0.11_Si_0.89_ film. Self–assembled Ge QDs are fabricated, and the hut-cluster or pyramid-like QDs can be clearly observed. The height distribution of Ge QDs is shown in Figure 1b and it can be fitted by two Gaussian peaks, which clearly demonstrate two types of Ge QDs. The extracted average heights (standard deviation) for the pyramid-like and the hut-cluster Ge QDs are 27.43 (4.40) and 38.25 (3.81) nm, respectively. While the density of the hut-cluster and pyramid-like QDs is ~9.76/μm^2^ and 11.96/μm^2^, respectively. These results are consistent with previous reports [15]. The Ge QDs layers were separated by 20 nm thick Ge_0.11_Si_0.89_ spacer layers, which were grown at 480 °C. Subsequently, the sample was capped by a 100 nm thick Ge_0.11_Si_0.89_ layer grown at 400 °C. Finally, a 100 nm thick Si layer was grown at the same temperature, which acted as the protection layer during the etching progress and then removed during KOH etching. For the fabricated unpatterned film sample, the broad peaks from 1250 nm to 1650 nm essentially originate from the Ge QDs embedded in the GeSi alloy film, as demonstrated in Figure A1a,b in Appendix A. In the unpatterned films, the PL spectra of Ge QDs can be decomposed into two typical Gaussian peaks NP and TO (Figure A1c). The NP and TO peaks were extracted as a function of the excitation power, and the blue shift of NP and TO peaks with power increasing can be clearly seen (Figure A1d).

Figure 2 schematically shows the manufacturing procedures for the GeSi hexagonal microdisk array. Firstly, PS microspheres in hexagonal lattice were self-assembled on the prepared GeSi films via a modified LB technique, as shown in Figure 2a. The size of PS microspheres was reduced by oxygen plasmons during a 70 s RIE with the power and gas flow values is 50 mW and 20 sccm, respectively (Figure 2b). Although the etching process of RIE is isotropic, the etching rate at contact points between PS spheres is much suppressed, leading to the formation of polymer sphere templates with well-defined hexagonal boundaries [26]. Then, the hexagonal microrods (MRs) were obtained by ICP with directly alternate cycle etching for 10 cycles with the C_4_F_8_ and SF_6_ gas (Figure 2c). During the ICP etching progress, SF_6_ gas was introduced to etch the exposed area of the PS template for 18 s and C_4_F_8_ gas was introduced to protect the side walls of the MRs for 10 s. Subsequently, the PS ball template was removed by tetrahydrofuran soaking of the sample (Figure 2d). Finally, the microdisk array shown in Figure 2e was realized by selectively Si etching in a 0.3 M KOH solution at 35 °C. Meanwhile, a circular microdisk array of comparable size was fabricated by replacing the direct cyclic etching in Figure 2c with a proportional mixture of C_4_F_8_ and SF_6_ gas.

The microdisk array sample is characterized by scanning electron microscopy (SEM) (Zeiss Sigma 300, Jena, Germany). The photoluminescence (PL) measurements are performed in a closed-cycle helium cryostat at the temperature of 17–300 K. The excitation source is a solid-state laser with λ_pump_ = 473  nm and the incident angle of the laser is ~50°. The ellipse laser spot on the sample surface has a size of about R~1.5 mm and the PL signal is collected by a lens with a collection angle of 20°. A monochromator (Omni–λ500, Zolix Instruments Co., Beijing, China) and an extended InGaAs photodetector with the standard lock-in technique are applied to measure and analyze the luminescence.

The optical resonance modes and the light field distributions in the hexagonal and circular microdisk are simulated by the FDTD method. As shown in Figure A2, we set the perfectly matched layers (PML) for the absorption boundary condition. The polarization orientation of the active medium is represented by three mutually orthogonal dipoles: radial (in-plane), azimuthal (in-plane), and along the *Z* axis (out of plane). Each individual hexagonal microdisk has a side length and thickness of 860 nm and 300 nm, respectively, and the circular microdisk has a side length and thickness of 900 nm and 300 nm. The distance between the FDTD boundary and the microdisk edge is 2 μm. The spectra of the dipole source are Gaussian shape with its width and wavelength center taken to be 1000 nm and 1500 nm, respectively. The dipole source was positioned 150 nm away from the upper and lower surfaces of the microdisk, which represent the QDs active region.

## 3. Result and Discussion

Figure 3 displays SEM images of the fabricated hexagonal microdisks. The side dimension and thickness of microdisks are about 860 and 300 nm, respectively, with a period of 2100 nm for the microdisk array. Figure 3a shows a top-view SEM image of the hexagonal microdisk array. The microdisks are hexagonal in shape and arranged in a hexagonal lattice with the microdisk angles opposite to each other. By changing the RIE etching time and the size of the PS spheres, the microdisk size and the microdisk array period can be adjusted. The GeSi hexagonal microdisks are produced and supported by Si nanorods, as shown in Figure 3b,c, by adjusting the concentration, temperature, and etching time of the KOH solution. The diameter of these Si nanorods is about 600 nm. As seen in Figure 3c, a stack of 5-layers Ge QDs separated by 20 nm thick Si spacers is embedded in hexagonal microdisks.

Figure 4a shows the PL spectra of an unpatterned film and a hexagonal microdisk array at an excitation power of 800 mW. The Ge QDs embedded in the GeSi alloy film are primarily responsible for the broad PL peak at around 1350–1650 nm of the unpatterned film, whereas the microdisk array exhibits several sharp resonant PL peaks. The integrated PL intensity from 1350 to 1650 nm of the microdisk array is about 2.5 times stronger than that of the unpatterned sample, which demonstrates that the presence of the cavity can significantly enhance the PL intensity of Ge QDs. Such an enhancement is attributed to the Purcell effect of microdisk and the improvement of extraction efficiency. The PL spectrum of the hexagonal microdisk array can be decomposed into seven Lorentz peaks (designated as H1–H7), as shown in the inset of Figure 4a. In addition, the wavelength of each fitting peak remains constant as excitation power increases as shown in Figure 4b. Therefore, we argue that these peaks are related to the cavity modes. The temperature-dependent PL spectra of the hexagonal microdisk array at 800 mW are shown in Figure 4c. Interestingly, substantially strong peaks can still be seen even at room temperature (T = 300 K, the inset in Figure 4c). The peak wavelength redshifts slightly when the temperature rises, as shown in Figure 4d. Such a redshift results from the change of the refractive index and the diameter of the microdisk with temperature.

To compare the luminescence between regular polygonal and typical circular microcavities with comparable sizes, we fabricate a similar array of circular microdisks using the same size PS spheres template. For the fabrication of the hexagonal microdisk, the SF_6_ and C_4_F_8_ gas are used separately for cyclically etching at a high power. Instead, the mixture gas of SF_6_ and C_4_F_8_ is used for etching at low power for the fabrication of the circular microdisks. Figure 5a shows the PL spectra of the circular microdisk and their spectrum fitting with eleven Lorentz peaks. The integrated PL intensity among 1350–1650 nm of the circular microdisk array is ~1.2 times stronger than that of the unpatterned film.

In general, when a quantum emitter (such as QDs) interacts with an optical resonator in a weakly coupled state, the enhancement factor F of the spontaneous emission rate is often given as below:(1)$$F=\frac{\Gamma }{\Gamma_{0}}=F_{p}\cdot \frac{{\left|\vec{u}\left({\vec{r}}_{em}\right)\cdot \vec{d}\right|}^{2}}{{\left|\vec{d}\right|}^{2}}\cdot \frac{1}{1+{\left(\frac{2Q}{\hbar \omega_{c}}\right)}^{2}\cdot {\left(\hbar \omega_{0}-\hbar \omega_{c}\right)}^{2}}$$
(2)$$F_{p}=\left(\frac{6\pi c^{3}}{n^{3}\omega^{3}}\right)\cdot \left(\frac{Q}{V_{mode}}\right)$$
where *Γ* and *Γ*_0_ are the SE rates of the QDs in a cavity and in free space, respectively, $\vec{u}\left({\vec{r}}_{em}\right)\cdot$ is the mode amplitude at the emitter position ${\vec{r}}_{em}$, $\vec{d}$ is the dipole moment, ω_0_ is the emitter oscillation frequency, ω*_c_* is the cavity mode frequency, *Q* is the quality factor, and *F_p_* is the maximum value of the Purcell factor [15]. From Equations (1) and (2), only the *Q*-factor cannot completely provide a description of the enhancement effect from the microdisk structure. The size uniformity and the *Q*-factor of the microdisks prepared by the NSL method appear to be unsatisfactory [15]. Furthermore, the blunt corners of the hexagonal microdisks lead to more light leakage and, thus, lower the *Q*-factor [28]. In order to effectively assess the enhancement effect of the microdisk structure, we employ a different method, known as the wavelength-dependent enhancement factor (*EF*), as follows:*EF*(*λ*) = *I*_disk_(*λ*)/*I*_film_(*λ*)(3)
where *I*_disk_(*λ*) and *I*_film_(*λ*) represent the PL intensity of the microdisk array and the flat film at wavelength *λ*, respectively. *EF*(*λ*) defined in Equation (3) is based on a direct comparison of experimental data, providing a direct description of the PL enhancement of Ge QDs luminescence in GeSi materials by microdisks structure. The *EF*s of the hexagonal microdisk and the circular microdisk array as a function of wavelength are shown in Figure 5c,d, respectively. Four *EF* peaks correspond to the wavelengths of peaks H2, H4, H6, and H7 are shown in Figure 4c with a maximum *EF* value is 20.4 for peak H6. Meanwhile, the circular microdisk also exhibits peaks C6, C8, and C11 with *EF* values of 14.9 and 14.0 for peaks C8 and C11, respectively, as shown in Figure 5a,b. These *EF* values are significantly larger than 1 (Figure 5d), which is attributable to the Purcell effect of the cavity mode. The *EF* values of the circular microdisks (C6, C8) are close to previous results of the GeSi circular microdisks [15]. It is found that the *EF* value 20.4 of the TE_5,1_ (H6) mode in the hexagonal microdisk is higher than those of all modes in the circular microdisk, indicating that the hexagonal microdisk structure has a unique enhancement for a certain mode. It is worth noting that peak TE_5,1_ (H6) has the largest full width at half maximum (FWHM) and thus the lowest *Q*-factor, implying that a direct description of the PL enhancement in Equation (3) is meaningful.

In order to illustrate the different enhancements of the hexagonal and circular microdisks, the TM and TE polarization modes in the photoluminescence spectra were identified successfully by the FDTD simulation at first. Figure 6a,b show the emission spectra from 1300 to 2000 nm of the hexagonal and circular microdisks, as well as the fitting curves for each resonant peak. The TM and TE modes are identified by green squares and black triangles, respectively, while the theoretical locations of the TM (TE) resonance peaks are marked by the vertical red (blue) dotted lines. A series of fundamental modes such as the TM_m,n_ modes (radial mode order n = 1, azimuthal mode order, 5 ≤ m ≤ 7) and TE_m,n_ modes (n = 1, 4 ≤ m ≤ 7) are visible in the hexagonal microdisk array, and TM_m,n_ modes (n = 1, 6 ≤ m ≤ 7) and TE_m,n_ modes (n = 1, 5 ≤ m ≤ 9) are visible in the circular microdisk array. Both TE and TM modes with high photoluminescence intensity exist on the short wavelength side, and high-order modes such as TM_6,2_ and TM_5,2_ are observed in the circular microdisks. The appearance of the high order means that more energy is required to compensate for the probable multi-mode competition in the circular microdisks. At the same time, with the increase in excitation power, the high-order mode will reduce the beam quality of the resonator. In this case, the hexagonal microdisk seems to have an advantage over the circular microdisk.

To further elucidate the experimental results, we conducted simulations and analyses of the optical fields corresponding to the modes of the *EF* peaks depicted in Figure 5c,d. Figure 6b exhibits the optical field distribution of the modes in the hexagonal and circular microdisks, including TE_6,1_ (H4), TE_5,1_ (H6), and TE_4,1_ (H7) modes for the hexagonal microdisk and TE_7,1_ (C6), TE_6,1_ (C8), and TE_5,1_ (C11) modes for the circular microdisk. The effective position matching between the optical field enhancement region and the Ge QDs can optimize the light–matter interaction, resulting in luminescence enhancement of the QDs in the microdisks. However, the position distribution of the self-assembled Ge QDs in the fabricated microdisks is random, so the statistical number of the Ge QDs in the field enhancement region is proportional to the area of the field enhancement region [14]. The area ratio *η* = *S*_e_/*S*_c_ is suggested to estimate the enhancement effect of PL for QDs by microdisks, where *S*_e_ is simply taken to be the area of the region in which the optical field intensity is larger than the half of the maximum value, Sc represents the area of the resonator cross-section. As shown in Figure 6d, the purple hexagons represent the η values of the modes for the hexagonal microdisks, while the orange circles represent those for the circular microdisks. From the perspective of light–matter interaction, we believe that a large value of *η* corresponds to a significant enhancement effect for the PL intensity of QDs in microdisks. As seen in Figure 6d, the TE_5,1_ mode exhibits the highest value of *η*, which could be the primary reason for the maximum *EF* value of 20.4 observed for the H6 mode in Figure 5c. Combined with the previous discussion, it can reasonably explain the unique enhancements of the hexagonal microdisks for H6 mode compared to the circular microdisks with comparable sizes.

## 4. Conclusions

In summary, periodically arranged hexagonal GeSi microdisk arrays with five Ge QDs layers embedded in the microdisk were fabricated via further structural modification based on the mature nanosphere lithography for the first time. The wavelength-dependent *EF* was extracted from the PL spectra of the hexagonal and circular microdisk arrays to analyze their emission enhancement. The *EF* value of 20.4 for TE_5,1_ mode in the hexagonal microdisk is higher than that of the circular microdisk with a comparable size. The unique enhancement for the TE_5,1_ mode in the hexagonal microdisk is comparatively analyzed through simulation of optical field distribution in the cavity.

## Figures and Tables

**Figure 1 nanomaterials-13-02553-f001:**
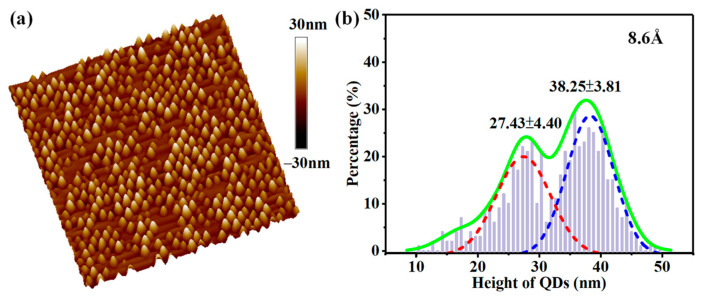
Surface morphology of Ge QDs on a GeSi alloy film on a Si (001) substrate. (**a**) AFM image (5 × 5 μm^2^) of Ge QDs after 8.6 Å Ge deposition at 520 °C on a 100 nm GeSi alloy film; (**b**) the height distribution of Ge QDs and the corresponding Gaussian fits.

**Figure 2 nanomaterials-13-02553-f002:**
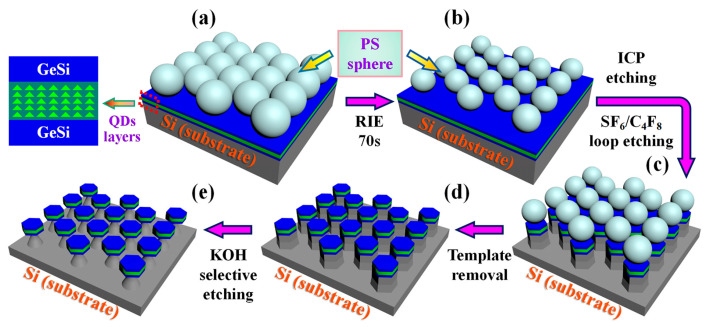
Schematic fabrication processes of Si-based GeSi hexagonal microdisk array embedded with Ge QDs. (**a**) Fabricating monolayer polystyrene (PS) microspheres template by the Langmuir–Blodgett (LB) technique; (**b**) etching the PS microspheres layer by the reactive ion etching (RIE); (**c**) direct loop etching for fabrication of microrods by the inductively coupled plasma (ICP); (**d**) removing the PS microspheres template; (**e**) wet isotropic etching of silicon by the KOH etchant.

**Figure 3 nanomaterials-13-02553-f003:**
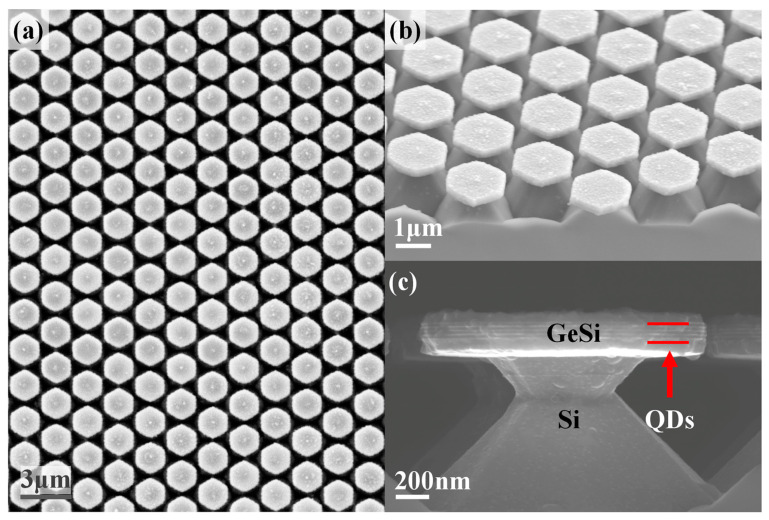
SEM images of the prepared GeSi hexagonal microdisk array. Three SEM images of the hexagonal GeSi microdisk array with embedded Ge QDs are shown: (**a**) the top view, (**b**) the oblique view, and (**c**) the side view.

**Figure 4 nanomaterials-13-02553-f004:**
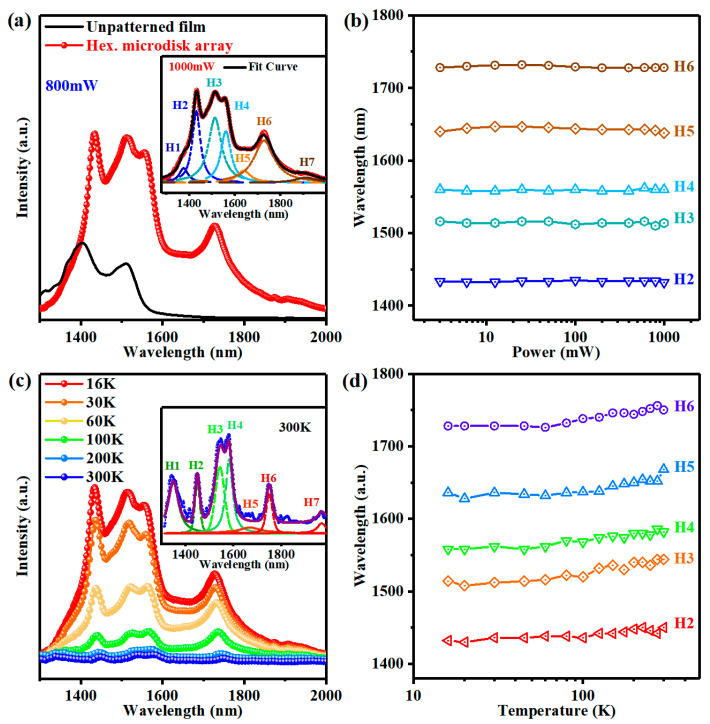
PL spectral results of the hexagonal GeSi microdisk array. (**a**) PL spectra of the hexagonal GeSi microdisk array and the unpatterned film both with self-assembled QDs at 16 K under an 800 mW excitation power. Inset: the PL spectra of the hexagonal GeSi microdisk array fitted by seven Lorentz peaks (labeled as H1–H7, dotted line). (**b**) The wavelengths of five selected peaks as a function of the excitation power (the weak peaks H1 and H7 are not included). (**c**) Temperature-dependent PL spectra of the GeSi hexagonal microdisk array with embedded Ge QDs. (Inset: PL spectra and Lorentz fitted peaks of the hexagonal microdisk array at 300 K). (**d**) The wavelengths of the five peaks as a function of the temperature under an excitation power of 800 mW.

**Figure 5 nanomaterials-13-02553-f005:**
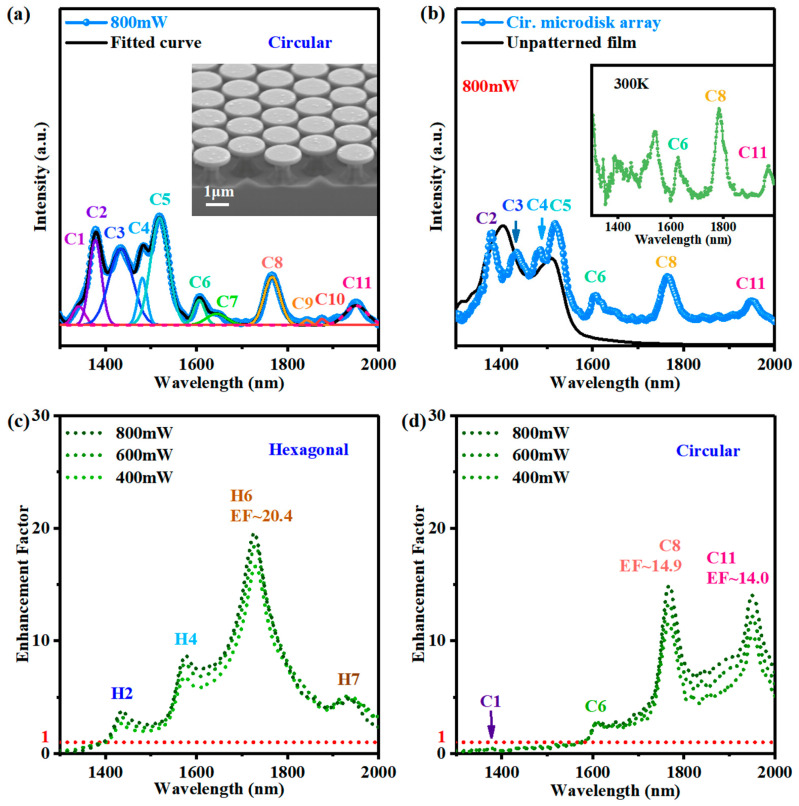
(**a**) Fitting the PL spectrum of the circular microdisk array at 800 mW with eleven Lorentz peaks. (Inset: the SEM image of the circular microdisk array); (**b**) the PL spectra of the unpatterned film and the circular microdisk array with a size similar to the hexagonal microdisks at 16 K and an excitation power of 800 mW. The power-dependent enhancement factor (*EF*) of (**c**) the hexagonal microdisk array and (**d**) the circular microdisk array as a function of wavelength for various excitation powers. The red dotted line in (**a**,**b**) indicates that *EF* = 1.

**Figure 6 nanomaterials-13-02553-f006:**
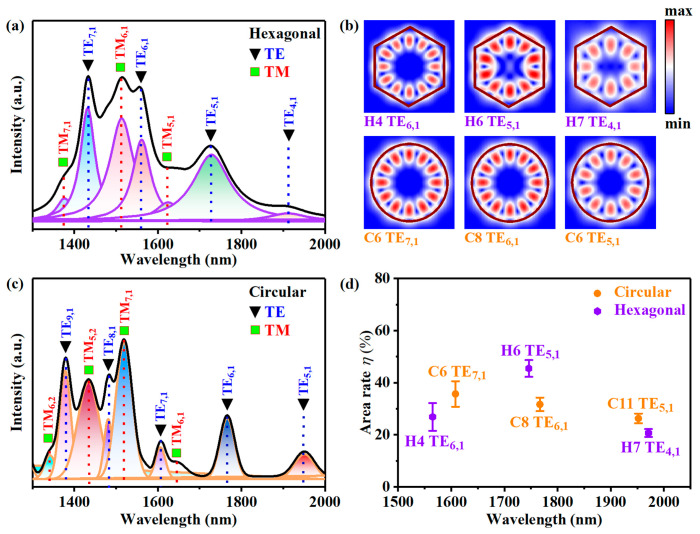
Photoluminescence spectra and fitting curves of mode peaks from (**a**) the hexagonal microdisk array and (**c**) circular microdisk array at excited power of 800 mW; black triangles and green squares are used to identify the transverse-magnetic (TM) and transverse-electric (TE) series mode peaks, respectively. (**b**) The simulated optical field intensity distribution of some modes that possess *EF* > 1. (**d**) The area ratio (*η*) of the area that the enhanced region exceeds half of the maximum optical field intensity value to the area of the resonator cross-section. The statistical results are plotted with error bars.

## Data Availability

Not applicable.

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
