# Peer review of "Unique Enhancement of the Whispering Gallery Mode in Hexagonal Microdisk Resonator Array with Embedded Ge Quantum Dots on Si"

_nanomaterials, 2023, doi:10.3390/nano13182553_

Round 1
Reviewer 1 Report
In this work the authors detail the fabrication of a GeSi hexagonal microdisk array and characterize its optical properties. The authors demonstrate that the use of hexagonal microdisks enhances their properties at certain wavelengths. In general the description of the research is quite good and shows the different structure property relationships. I believe that this manuscript should be accepted, potentially with some minor revisions.
The SEM images of the microdisk arrays are quite impressive and show that the fabrication technique is highly precise. The germanium QDs are described in the schematic in Figure 1, and the authors claim that they were deposited by S-K growth during fabrication. Unfortunately, there are no microscopy images of the individual quantum dots, which is understandable considering that they could not be imaged after fabrication is completed due to their encapsulation. Do the authors have images of the grown quantum dots if they halt the fabrication process? I believe that images of the quantum dots would help illustrate the quality of the fabrication.
Figure 5d is the most important figure in the manuscript, as it shows the enhancement with the different structures that were fabricated. Unfortunately there seems to be only small differences between the hexagonal and circular microdisk enhancements at given wavelengths. The authors should estimate the error on these measurements and plot the points with error bars so the reader can determine if these are statistically significant enhancements.
Also, 21 citations seems like a very small number for this manuscript. I believe the authors could do a more thorough job connecting their work to the literature, and this number of citations could easily be doubled.
Very minor grammatical issues noted, proofreading by a native speaker with a scientific background would likely fix any issues.
Reviewer 2 Report
The authors fabricated hexagonal microdisks embedded with Ge QDs by nanosphere lithography and investigated the enhancement of the incident optical field by WGM and its effect on the resonant emission of the QDs. The results show a 20-fold enhancement of the emission, and a comparison with circular microdisk arrays reveals the usefulness of interesting resonance modes due to the polygonal geometry. These results also show good agreement with FDTD simulations. Since this is a very interesting paper and a significant contribution to the research area, I consider it acceptable for publication after the following modifications are addressed.
1) The influence of optical absorption must be taken into account when considering the EF, but it is not stated how this has been taken into account. It is assumed that there is a difference in excitation light absorption between the film and the array.
2) H6 emission peak is not seen in the film. An excitation spectrum measurement is needed to confirm that this is indeed the emission from Ge QDs.
Reviewer 3 Report
The authors fabricated a periodically arranged hexagonal GeSi microdisks arrays with five Ge quantum dot (QD) layers embedded in the microdisk. The wavelength–dependent enhancement factor (EF) was extracted from the photoluminescence (PL) spectra of the hexagonal and circular microdisk arrays for investigating their emission enhancement. It is demonstrated that the EF value of 20.4 for TE5,1 mode in the hexagonal microdisk is higher than that of the circular microdisk with a comparable size. The enhancement for this TE5,1 mode in the hexagonal microdisk was studied in numerical simulation. The results are novel and interesting.
I can suggest to the authors to discuss Q-factors of the prepared samples and their limitations. This can be useful for readers.
I recommend to enlarge figures 3,4,5. It is hard to read text in them.
In formulas (1,2) solid black circle should be replaced with dot.
